# Evaluation of Disability Progression in Multiple Sclerosis via Magnetic-Resonance-Based Deep Learning Techniques

**DOI:** 10.3390/ijms231810651

**Published:** 2022-09-13

**Authors:** Alessandro Taloni, Francis Allen Farrelly, Giuseppe Pontillo, Nikolaos Petsas, Costanza Giannì, Serena Ruggieri, Maria Petracca, Arturo Brunetti, Carlo Pozzilli, Patrizia Pantano, Silvia Tommasin

**Affiliations:** 1Institute for Complex Systems, National Research Council (ISC-CNR), 00185 Rome, Italy; 2Department of Advanced Biomedical Sciences, Federico II University of Naples, 80131 Naples, Italy; 3Department of Electrical Engineering and Information Technology, Federico II University of Naples, 80125 Naples, Italy; 4Department of Radiology, IRCCS NEUROMED, 86077 Pozzilli, Italy; 5Department of Human Neurosciences, Sapienza University of Rome, 00185 Rome, Italy; 6Neuroimmunology Unit, IRCSS Fondazione Santa Lucia, 00179 Rome, Italy; 7Department of Neuroscience, Reproductive Sciences and Odontostomatology, Federico II University of Naples, 80131 Naples, Italy

**Keywords:** deep learning, disability, magnetic resonance imaging, multiple sclerosis, neuroimaging

## Abstract

Short-term disability progression was predicted from a baseline evaluation in patients with multiple sclerosis (MS) using their three-dimensional T1-weighted (3DT1) magnetic resonance images (MRI). One-hundred-and-eighty-one subjects diagnosed with MS underwent 3T-MRI and were followed up for two to six years at two sites, with disability progression defined according to the expanded-disability-status-scale (EDSS) increment at the follow-up. The patients’ 3DT1 images were bias-corrected, brain-extracted, registered onto MNI space, and divided into slices along coronal, sagittal, and axial projections. Deep learning image classification models were applied on slices and devised as ResNet50 fine-tuned adaptations at first on a large independent dataset and secondly on the study sample. The final classifiers’ performance was evaluated via the area under the curve (AUC) of the false versus true positive diagram. Each model was also tested against its null model, obtained by reshuffling patients’ labels in the training set. Informative areas were found by intersecting slices corresponding to models fulfilling the disability progression prediction criteria. At follow-up, 34% of patients had disability progression. Five coronal and five sagittal slices had one classifier surviving the AUC evaluation and null test and predicted disability progression (AUC > 0.72 and AUC > 0.81, respectively). Likewise, fifteen combinations of classifiers and axial slices predicted disability progression in patients (AUC > 0.69). Informative areas were the frontal areas, mainly within the grey matter. Briefly, 3DT1 images may give hints on disability progression in MS patients, exploiting the information hidden in the MRI of specific areas of the brain.

## 1. Introduction

Disability progression is highly heterogeneous in all forms of multiple sclerosis (MS) [1]. Various studies have evidenced that the autoimmune response towards the central nervous system leads to chronic inflammation, axonal degeneration, and remyelination phenomena and causes disease progression [2]. However, since it is not yet clear how these phenomena interact with each other over the course of the disease, clinical outcomes are highly variable and difficult to predict.

A multiplicity of factors may influence the rate of disability progression, including genetic and environmental factors, treatment, age at onset, and disease duration, as well as the severity of tissue damage [3]. Further, measures that are significantly correlated with the disease course, population-wise, do not necessarily predict the clinical outcome of individual subjects with absolute certainty. Therefore, although various prognostic markers are now available, and magnetic resonance imaging (MRI) has become an essential tool for diagnosis, treatment decisions, and disease monitoring, it remains often difficult to decide the best therapeutic strategy for a given patient due to the uncertainty related to his/her individual progression [4].

As the mechanisms underlying disability progression are complex, the identification of reliable predictors would require elaborated models, able to identify the relative weight of various factors, considered alone or in association. In the past few years, deep learning (DL) neural networks represented a breakthrough in the domain of image classification and prediction, proving their power in the analysis of large image datasets along with their ability to perform sophisticated visual recognition tasks [5]. So far, DL classifiers for MS have exhibited promising results in the image segmentation of brain structures and tissues [6,7,8,9,10,11,12]. Further, DL artificial neural networks have shown a good ability to correctly classify MS against several white matter disorders or other MS-mimics, exploiting T2-weighted (T2w) and T1-weighted (T1w) brain MRI scans [13]. Furthermore, when combining T1w with myelin maps in multimodal MR images, DL networks were shown to potentially improve the early-stage detection of MS disease [14].

On the basis of the promising results in image segmentation and classification, we tested the DL ability to predict disability progression in MS. Indeed, to predict clinical status and worsening in patients with MS, artificial intelligence algorithms were applied either on MRI [15,16,17,18] and/or clinical data [19,20]. In this pilot study, we hypothesized that the disability progression, as evaluated via the Expanded Disability Status Scale (EDSS) after 4 years from the baseline visit, may be predicted from structural MRI images acquired at baseline. Thus, we aimed at devising DL artificial neural networks to classify patients at baseline according to their 4-year follow-up status (progressed versus not progressed) on one side and to identify the brain areas that may be good indicators of the disability worsening on the other.

## 2. Results

### 2.1. EDSS-Discriminators

We analyzed the statistics of the 100 AUC values, obtained by considering model predictions for any slice individually (see Figure 1, top panels).

Considering the coronal and sagittal slices, generally, the medians of AUC distribution per slice were close to 0.5, the 25% and 75% percentiles were close to AUC = 0.4 and AUC = 0.6, and every slab obtained from three to seven outliers, thus the best performing models, towards AUC = 1. On the contrary, slabs of axial slices obtained a larger number of outliers towards AUC = 1, i.e., more than six, even if keeping median AUC = 0.5 for the majority of the slices.

We also performed the statistical analysis of the 100 AUC values, obtained by considering the predictions for all MRI slices in each of the four slabs per projection plane. The AUC distributions are shown in Figure 1 (bottom panels). They all exhibited a pronounced peak around ~0.5 and a closer inspection revealed that the mean values and medians coincided in all cases (0.01 was the maximum difference), standard deviations ranged from 0.034 to 0.064, and the 75th percentile was never larger than 0.54. The outliers of the coronal and sagittal models were 0 for all slabs, with the exception of coronal slab 2, which had two outlier models, and sagittal slab 1, which had one outlier model. The outliers of models built on axial slabs were one for all slabs, with the exception of slab 1, which had four outlier models. Given that half of the investigated slabs had zero outliers, thus showing no class separation capacity whatsoever, and whole structures of the brain would be missed by the following analysis, comparison with the null EDSS-Ds was not performed for slab-based predictions.

### 2.2. Comparison with Null EDSS-Ds

The null EDSS-D was used as a term of comparison to verify whether each classifier displayed some non-trivial features. However, an appropriate null model would behave in accordance with the reasonable null hypothesis of having no discriminatory efficiency, i.e., AUC = 0.5. Hence, models with a null AUC ∈ [0.5 ± 0.0139] and corresponding statistically different real AUCs according to DeLong’s test were selected if the real AUC was towards 1. Figure 2 displays the slice-classifier analysis, thus the slices whose AUC fulfilled the above-mentioned criteria of null AUC ∈ [0.5 ± 0.0139] significantly different from the relative real AUC, which was close to 1.

As displayed in Figure 2, five coronal slices had one classifier fulfilling the criteria and thus predicting disability progression. These slices were located in slab 4 and AUC ranges within [0.72–0.93]. Similarly, five sagittal slices had one fulfilling classifier, with AUC ranges within [0.81–0.90].

Lastly, 13 axial slices had at least one fulfilling classifier and were preferably located in slab 4. Specifically, 15 combinations of 12 unique classifiers and 13 unique axial slices resulted to fulfill the criteria, with the AUC ranging within [0.69–0.86].

In general, good-performing classifiers, i.e., models which met the aforementioned criteria of admissibility, were different in each slice, with the only exception of two cases. Two classifiers predicted disability progression on two axial slices covering either the thalamus, basal ganglia, ventricles, and more, or supplementary motor area (slab 2, z = 39; slab 4, z = 70, Figure 2).

### 2.3. Three-Dimensional Representation

By the intersection of significant slices, we obtained 375 significant voxels, whose location was displayed as an area in the three-dimensional map in Figure 3. Inspecting the voxels’ location, we obtained an overview of the areas that may contain most of the information useful to predict disability progression. These areas were located on the frontal pole, mainly within the grey matter.

## 3. Discussion

From the application of DL classifiers on slices derived by 3DT1 preprocessed images, we extracted and exploited features hidden in the MRI images to identify patients who would experience disability progression. We found that the majority of the information useful in predicting disability progression resides in specific brain areas. This conclusion prevented us from reaching an integration strategy of all the results; thus, to obtain a unique classification for each brain image, but unveiled that 3DT1 images may offer hints on the prediction of disability progression in MS. As well, it suggested that a slice-based approach may disclose many features that would be lost by using a convolutive approach to reduce image dimensionality.

The prediction of disability progression in MS via machine learning tools is a topic under discussion and so far authors have found that features extracted from brain MRI have good discriminatory power (e.g., [16,21,22,23]). Indeed, even though disability progression may be predicted on clinical data alone [24], MRI data may be of use because MRI is sensitive in detecting pathological signs. With our approach, we aimed at investigating, with DL algorithms, the information hidden within 3DT1 MRIs, which may be used in clinical practice, in order to predict disability progression avoiding any elaborate MRI processing and manipulation.

### 3.1. Two-Step Training of Pre-Trained Deep Neural Network

One of the DL fundamental requirements is the availability of a large number of labeled images in the training. Unfortunately, the ease of access to a large labeled medical images dataset is a problematic issue for a series of ethical and practical reasons: patient confidentiality concerns, a lack of standardized protocols, and the expensive nature of annotation, among others [7]. Several strategies have been crafted to overcome this drawback, including fine-tuning a pretrained network optimized on large data in a different domain (i.e., transfer learning [25,26,27,28]), using a pretrained network in the same domain [29] and data augmentation [30].

Our protocol has been to fine-tune the pre-trained ResNet50 model in subsequent steps, making the classifier progressively learn the brain features to achieve the binary discriminating performance which was required as a final goal. Since the pre-trained ResNet50 is a convolutional neural network trained on more than a million images not related to MRI, the first step was to adapt ResNet50 to MRI. The first training applied the neural network to more than a thousand unlabeled MRIs in order to allow it to recognize structures within MRI, thus, to correctly locate slices in slabs.

After the first application onto MRI, the neural network model was suited for the second training on the labeled MRI aiming at correctly classifying patients with stable or progressed disability.

### 3.2. Models Built on Either Single Slice or Slice Combination

Since anatomic sections of the brain may be very different and do not include the same amount of morphological affine structures, convolution or other methods of data combination may cause information loss in the data pool. Hence, in order to keep as much information as possible, we avoided slices combination, which is the most common way to analyze three-dimensional data with DL (e.g., [23]). Indeed, in our study, classifiers built on entire slabs, thus entire brain sections, showed no average discriminative power.

Although we built models based on both slabs and single slices, we hypothesized that some models might perform at best on a slice level as the information, if any, might be hidden in some specific area, smaller than the slab size. The single slice analysis showed that, at least in the case of coronal and axial projections, a number of classifiers showed good discriminative power. This might be interpreted as a potentially relevant statistical signal about the precursory information hidden in some slices. However, the extremely noisy AUC distribution profile could also imply that better performances were ascribable to statistical fluctuations due to the limited number of patients in the training set. To settle this issue, we separated the real signal from the inherent noise comparing each model AUC against its ground truth, given by the AUC of the null model.

### 3.3. 3-DT1 Images to Predict Disability Progression in MS

In this pilot study, we found only a few (two) classifiers able to predict disability progression in more than one slice. Such a small number of classifiers may be due to the small sample size or to the use of 3DT1 images alone. Indeed, DL classifiers extract features directly by the image, building the identification on voxel values and not on anatomical characteristics. A recent study has addressed the issue of decision-making on T2-weighted images, finding that individual lesions, lesion location, and non-lesional white matter or gray matter areas are informative features [31]. As a matter of fact, white and grey matter atrophy and damage, e.g., black holes, may be clearly extracted by 3DT1 MRI images, while less destructive lesions are not as clearly visible as in T2-weighted MRI images: in 3DT1 images, the use of DL may uncover hidden features related to lesions that would not be detected with other techniques. Hence, the use of 3DT1 images in predicting MS-related disability progression may be as relevant as T2-weighted images, which are universally recognized to offer the ground truth lesion maps [32], and are the most used in DL studies on MS (e.g., [10,11,23,31,33]).

### 3.4. Anatomical Structures Relevant in Predicting Disability Progression

Axial projection resulted in being the most informative in predicting disability progression, since the largest number of efficient classifiers were developed on forebrain axial slices. Indeed, these axial slices covered areas that are known to be affected in multiple sclerosis and whose damage is clinically relevant [34,35,36], including the third and lateral ventricles, whose enlargement is an indirect measure of brain tissue loss. Analogously, coronal and sagittal slices may include the right amount of information to build on them efficient classifiers, especially if spanning disease-affected areas, such as frontal and parietal lobes, somatosensory cortex, and cerebellum [37,38,39]. We expect to overcome this failure by increasing the sample of tested patients.

However, the three-dimensional analysis showed the confluence of crossing points of projection-wise, discriminative slices and reiterated the relevance of frontal areas, mainly within the grey matter, in predicting disability progression [36].

### 3.5. Model Generalization

Patients recruited in the two sites had significantly different clinical characteristics and span a wide range of EDSS at baseline. Indeed, this EDSS range resembles the patients’ population, and disability progression was demonstrated to follow a two-stage process with a pivotal point at an EDSS equal to three [40]. These differences, added to the difference of the MRI tomographs, granted generalizability of the obtained results [41].

### 3.6. Limits

This study was pilot research intended to test the feasibility of predicting disability progression from the preprocessed structural images of patients with MS, therefore the small sample size will be incremented in future development. Increasing the sample size will allow to build a larger validation set and to introduce a test set. EDSS was used to evaluate disability in patients, and, as it is assigned by clinicians, it may suffer from bias due to the operator and influence the performance of the models. However, EDSS is the most widely used clinical outcome measure for disability evaluation [42] and was evaluated by neurologists with long-lasting experience. Additionally, as a development of the study, we will test the combination of 3DT1 with T2-weighted images, since the latter are considered the ground truth for lesion recognition. Further, the inclusion in the study of contrast-enhanced MRIs acquired within 4 years, thus at baseline and follow-up, would provide strong evidence to show the validity of the model in predicting progressed lesions. Finally, in order to allow the methodology to be appealing for clinical practice, a next step may be to skip the preprocessing step and classify the very raw images of patients with the models built with the presented two-step training algorithms.

## 4. Methods and Materials

### 4.1. Patients Selection and Disability Assessment

One-hundred-and-eighty-one patients with MS were recruited by two centers: the Department of Human Neuroscience of Sapienza University (Site 1) and the MS Center of the Federico II University (Site 2). Study protocols were approved as appropriate by the ethical committees of both Policlinico Umberto I/Sapienza University (Rome, Italy, Site 1) and the Biomedical Activities “Carlo Romano” of Federico II University (Naples, Italy, Site 2). All subjects provided written informed consent.

Patients with MS were retrospectively selected from the databases of the two sites, according to the following selection criteria: diagnosis of MS according to the McDonald’s criteria [43,44]; age between 18 and 70 years; baseline clinical assessment and MRI examination not more than one month apart; absence of relapses and/or steroid treatment in the 30 days preceding the MRI; clinical follow-up available after 2 to 6 years from the MRI examination.

At baseline and follow-up, patients were examined by neurologists with long-lasting expertise in MS care (10 to 30 years), who assessed their clinical disability status. Specifically, EDSS scoring was performed by certified neurologists (neurostatus.net). The EDSS score expresses the disability status of MS patients on a scale from 0 to 10 [45]. After the follow-up examination, disability progression was defined as a 1.5-point increase for patients with a baseline EDSS of 0, 1 point for scores from 1.0 to 5.0, and 0.5 points for scores equal to or higher than 5.5 [46].

### 4.2. Population

The average and standard deviations of demographics and clinical data for both Site 1 and Site 2 separately, as well as for the entire sample built from the two sites, are reported in Table 1. In total, 163 over 181 (90%) patients participated in another study, and their structural characteristics are reported in Tommasin et al. 2021. At the follow-up examination, disease progression was observed in 62 patients (34% of the sample) whose EDSS medians and ranges at baseline and at follow-up were, respectively, 3.5 [0.0–7.0] and 4.5 [1.5–7.5], while in the remaining 119 patients, the EDSS remained stable at 3.0 [0.0–7.5]. Disability progression was confirmed after 3 months. In the time interval between baseline and follow-up, 80 patients did not switch therapy, 32 patients changed therapy but not treatment line, and 69 patients changed treatment line. Moreover, 65 patients in Site 1 and 52 patients in Site 2 had no relapse, while 20 patients in Site 1 and 24 patients in Site 2 had 1 to 5 relapses. Clinical characteristics are described in Table 1.

### 4.3. Magnetic Resonance Imaging

Baseline MRI acquisitions presented the following features: Site 1 hosted a 3T Verio Siemens scanner, equipped with a 12-channel coil for parallel imaging and performed a high-resolution, three-dimensional, T1-weighted (3DT1) MPRAGE with 176 1-mm–thick sagittal sections (TR = 1900 ms, TE = 2.93 ms, TI = 900 ms, flip angle = 9°, matrix = 256 × 256, FOV = 260 mm^2^); Site 2 hosted a Trio Siemens, equipped with an 8-channel head coil, and performed a 3DT1 MPRAGE with 176 1-mm–thick sagittal sections (TR = 2500 ms, TE = 2.80 ms, TI = 900 ms, flip angle = 9°, matrix = 256 × 256, FOV = 256 mm^2^).

The 3DT1 images were corrected for field inhomogeneity, thus the spatial variations of image intensity, brain-extracted, and registered onto standard space via linear registration, as implemented in the FMRIB Software Library version 6.0 (Wellcome Centre for Integrative Neuroimaging, Analysis Group, University of Oxford, Oxford, UK, https://fsl.fmrib.ox.ac.uk/fsl/fslwiki, accessed on 1 December 2018). The 3DT1 images were registered onto the Montreal Neurological Institute (MNI) template [47], which is the most commonly used template in MRI research and has allowed comparisons among studies performed by several institutes in recent decades. After voxel intensity normalization to 255 (the highest value of the RGB color coding), the 3DT1 images were cut in slices along the coronal, sagittal, and axial projections following the standard planes parallel to the floor of the IV ventricle, to the medial plane and to the bicommissural line, respectively. Slices were selected for the analysis if they included at least 10% of the non-zero voxels. Each slice image was exported from the nifti format to jpg. Jpgs of slice images on each of the three projections, e.g., coronal, sagittal, and axial, were used to build DL classifiers.

### 4.4. Slab Selection

DL algorithms extract hidden features from images; therefore, to push towards the best performance, we built classifiers on groups of slices covering areas with similar histological organization as much as possible, e.g., keeping the majority of the cerebellum or the basal ganglia disentangled by the majority of the cortex. Therefore, for each projection, the slices were grouped in four contiguous slabs accounting for definite anatomical structures, covering roughly the same number of slices. The sagittal and axial projection slabs were devisable with the same number of slices. On the contrary, a different number of slices were grouped in coronal slabs to account for the brain anatomy.

Coronal projection included 79 slices ranging from y = 17 to y = 95 in standard MNI space. They were grouped into two posterior (1–2) and two anterior (3–4) slabs with respect to the central sulcus, as displayed in Figure 4.

Sixty-four slices were included on the sagittal projection ranging from x = 14 to x = 77, two right and two left slabs with respect to the midline, as in Figure 5.

Lastly, axial projection included 62 slices, ranging from z = 12 to z = 73, two below (1–2) and two above (3–4) slabs with respect to the posterior commissure, as in Figure 6.

### 4.5. Modelling Strategy

We have selected ResNet50 as the DL network backbone for the purposes of this study. This choice was motivated both by the excellent performance shown by ResNets50 in numerous computer vision tasks, as well as because it could efficiently run on the available hardware resources. The overall approach was to adapt the original model via a two-step training procedure (see Figure 7).

**First training: Brain Scan Optimized Model.** The first step consisted of fine-tuning the off-the-shelf ResNets50 network on a large set of non-labeled brain-extracted MRIs. The first step’s aim was to allow the network to learn the brain basic structural concepts in MRI and to classify images by location in the four predefined slabs in the three spatial projections. This training was independently performed on brain-preprocessed 3DT1 images of 1833 subjects obtained from public repositories and provided three different Brain Scan Optimized Models (BSOM), one for each of the three slice projections (coronal, sagittal, and axial) (Figure 4, top). The preprocessing of the public repository images was performed following the same procedure as the study images, thus images were corrected for field inhomogeneity, brain-extracted, and registered onto MNI space via linear registration.

**Second training: EDSS-Discriminator.** The second training step was performed on the set of disability-progression (EDSS) labeled MRI scans (Figure 4, bottom). Each of the three intermediate BSOM models specific to a projection was fine-tuned to a specific slab in order to create a total of twelve possible types of binary discriminators tailored to the determination of EDSS progression.

In order to assess the statistical significance of the results, we generated 100 distinct discriminative binary models due to the limited number of EDSS-labeled images in our database. Hence, a set of 100 distinct EDSS-Discriminators (EDSS-D) were trained in each of the three projection planes (coronal, axial, sagittal) divided into four slabs, resulting in a total of 1200 binary discriminators.

**EDSS-D performance and comparison with null model.** The performance of the EDSS-D discriminatory power was assessed by calculating the area under curve (AUC) for each brain slice in the validation set. Moreover, a proper control was implemented by comparing the AUC value with the corresponding value arising from a null model, defined as the ground truth.

Each step is extensively described in the following paragraphs.

#### 4.5.1. 1st Training Step: BSOM

We took unlabeled independent MRI 3DT1 images of 1833 subjects collected from public repositories, such as the Parkinson’s Progression Markers Initiative (938 images, PPMI, https://www.ppmi-info.org/, accessed on 1 April 2020) and Alzheimer’s Disease Neuroimaging Initiative (554 images, ADNI, https://adni.loni.usc.edu/, accessed on 1 April 2020), and from 341 more 3DT1 images of both patients with MS and healthy subjects obtained from Site 1 database. Patients with MS, whose 3DT1 image was included in the first training step, did not fulfill the inclusion criteria for this study and therefore could not be included in the study sample. We considered separately the coronal, axial, and sagittal projections. Slices belonging to each projection were classified into one of the four slabs via hierarchy topological feature extraction using ResNet50 architecture (see Figure 4, Step 1), implemented via the open-source Keras library with a Tensorflow backend. Four-category classification was formed by replacing the built-in ResNet50 top layer with a custom one. All three models showed the best performance by adopting Adam as the optimizer; the learning rates were set as 10^−3^ (coronal projection) and 10^−4^ (sagittal and axial projections). In order to meet the dimension requirement of ResNet50, the slice images were upsampled to 224 × 224 pixels and preprocessed using a homemade code to maximize the contrast and cause the image details to be more recognizable (see Supplementary Material Appendix A). Moreover, standard data augmentation was applied to the original batch images using Keras’ ImageDataGenerator class. Starting from pre-trained coefficients from the Imagenet dataset, the best classifiers performance was achieved by fine-tuning ResNet50 in two stages: at first, by freezing all layers but the top, and subsequently unfreezing all the layers but the first 133, corresponding to the first four convolutional blocks of ResNet50. To evaluate the models’ confusion matrix, the images distinguished in the three planes, e.g., coronal, sagittal, and axial, were devised in training and validation sets as the 90% and the 10% for each plan. The classifiers validation accuracy was evaluated at 97% for the coronal and axial models and 85% for sagittal. The resulting three BSOMs, each specified for the classification of images belonging to the specific projection plane, were then optimized in the second training step.

#### 4.5.2. 2nd Training Step: EDSS-Discriminators

Patients who displayed disability progression at the follow-up from the baseline visit were labeled with 1 (positive cases), and stable patients were labeled with 0 (negative cases). We created 100 training sets, different but not independent among each other, e.g., a patient could belong to more than one training set. This choice was decided upon to build a statistical ensemble of distinct but not independent replicas, over which DL models were tested. Patients belonging to training and validation sets were randomly selected, leaving unaltered the proportion between negative and positive cases with respect to the entire original sample. Each training set included 90% of the total patients: 163 patients subdivided into 107 negatives and 56 positives. Validation sets were then composed of 18 patients and stratified in order to be 12 negatives and 6 positives. The slices composing the 100 training (and validation) sets were additionally subdivided into four groups according to the slabs partitioning (see Figure 8, left panel).

Implementing binary classification required the modification of the previously trained BSOMs (i.e., built on slices parallel to the sagittal, coronal, or axial projection planes) by the substitution of a single unit in the last dense layer (Figure 7, Step 2). The three discriminators were fine-tuned separately on the ensemble of 100 training sets for each slab. In total, we scrutinized 100 × 3 × 4 different binary EDSS-Ds: 100 models per brain slab (four slabs for each projection plane). We invariably chose Adam as the optimizer. The best learning rate and number of epochs were preliminarily evaluated for each discriminator. As in the Step 1 training process, batch MRIs were expanded and subjected to data augmentation compatible with the Keras’ ImageDataGenerator class.

#### 4.5.3. AUC Statistical Analysis

Since the labeled MRI datasets were biased towards negative cases, we preferred the area under curve (AUC) over the accuracy as the efficacy metric for binary classification. AUC is in fact often preferred over accuracy for binary classification on highly unbalanced problems. In our specific case, indeed, negative patients are on average 65% of the total both in the training and validation sets, yielding an accuracy of 0.7 for any model. Therefore, models appeared to be representative but not discriminative, suggesting that we were overfitting to a single class, namely the negative class of patients. The AUC was calculated starting from the receiver operating characteristic (ROC) curve. The ROC curve was obtained by plotting the true positive rate (TPR), or sensitivity, against the false positive rate (FPR), or 1—specificity, at various threshold settings. We considered well-performing models those with an AUC close to 1.

The statistical analysis of the AUCs obtained from the different models could be created per slab or single slice. Thus, we took two approaches: one classified labeled MRI per slice and the other classified the MRI images per slab, thus folding all slices within each slab.

At first, slice-based AUC analysis aimed at evaluating the models’ binary selective performance at the single slice level. In this case, the models’ predictions were analyzed considering the slices individually. TPR consisted of the number of true positive patients that were correctly identified upon slice classification, divided by six, i.e., the number of positively labeled patients in the validation dataset. Slice FPR, instead, was the ratio between the number of false positive patients over the total number of negative patients in the validation set, i.e., 12. In this way, we obtained as many AUCs as many models per slice per projection. Considering the distribution of the hundred AUCs obtained for each slice, the median, 25th and 75th interquartile and outliers, defined as points at 1.5 times the interquartile range, were calculated. Outliers were considered best-performing models, if they reached an AUC significantly higher than the majority of the models built on the same slice.

For the second approach, i.e., considering all the MRIs in a slab, the performance of each binary classification model was assessed by evaluating the predictions for all slices within that slab as a whole. Thus, to evaluate each slab’s performance, the training dataset was comprised of the number of subjects (12) times the number of slices included in that slab, and the validation dataset was comprised of the number of subjects (6) times the number of slices included in that slab. Thus, the TPR was the fraction between the total number of truly positive predicted slices and the number of positive slices within each slab, and FPR was the total number of false positive slice predictions in a slab, divided by all negative slices in the same slab. The statistical analysis was performed on the average properties of the AUC in each slab. In this way, we obtained four AUCs for each projection and each of the hundred models.

#### 4.5.4. Null EDSS-Discriminators

A corresponding null model was assigned to each of the EDSS-Ds. A null model can be considered as the ground truth for its specific DL model and was defined as follows. The EDSS labels characterizing the images present in each training set were randomly shuffled, keeping the numbers of positive and negative patients unvaried. The validation sets and the other parameters of each DL classifier (learning rate, optimizer, number of epochs) remained unaltered (see Figure 8, right panel).

The binary class separation power of any null model was determined by its AUC, in analogy to the real case. Only those null classifiers with no binary classification power were considered, i.e., with an AUC ∈ [0.5 ± δ]. The systematic error δ in the AUC calculation was determined by the ROC stepwise form. On the ROC, the *x*-axis (FPR) discrete unit was 1/12, while in the *y*-axis, the sensitivity could be incremented (or decreased) by units of 1/6. Hence, the minimal AUC variation was equal to δ = 1/(6 × 12) = 0.0139. Models whose null classifier had AUC > |0.5 ± δ| were excluded from the following analysis.

Once a null model was selected, it had to be assessed whether the corresponding original model had a statistically different AUC from it. Three possible situations might occur (Figure 9): the real classifier AUC could be i. larger, ii. smaller, or iii. not significantly different from the AUC of the null classifier. We adopted DeLong’s test [48,49] to obtain a p-value to evaluate whether real and null models had significantly different AUCs. We used the pROC package in R [50] to calculate the Z score, namely the quantitative indicator of the AUC difference between the two classifiers. Under the assumption that Z was distributed according to the standard normal distribution, if the value of Z was such that Z > 1.96, it was reasonable to consider the two models statistically distinct, within the significance level *p* < 0.05. Hence, we selected only those models satisfying this condition.

#### 4.5.5. Three-Dimensional Representation

To have a 3D topological view of the brain areas prone to the disability worsening, we built a map of significant voxels to superimpose to the brain volume in MNI space. Significant voxels were identified by the intersection of slices along the three projections, with at least a classifier satisfying the aforementioned conditions. Each slice was characterized by its coordinate on the axis perpendicular to the projection plane (e.g., a coordinate on the *z*-axis for each axial slice), therefore the intersection of three slices, each on a plane, resulted in a complete set of coordinates identifying a voxel. The total number of significant voxels was the product of the numbers of significant slices on the coronal, sagittal, and axial planes.

## 5. Conclusions

The aim of this study was to develop an automated DL-based tool to predict disability progression via the EDSS after 4 years from the baseline visit, from 3DT1 images acquired at baseline. The results showed that not all of the slices building the 3DT1 images contain information useful to predict disability progression, at least with the relatively small size of the investigated sample, and therefore we could not develop a final integration strategy. Nonetheless, to investigate the whole information stored in 3DT1 images, avoiding convolutive steps to reduce image dimensionality may be the way to not to lose any relevant feature. Indeed, this study shows how even 3DT1 images may host hidden information about disability progression, especially due to the imaging representation of specific areas. Ultimately, this study may be a first step in the development of an automated tool for disability progression prediction.

## Figures and Tables

**Figure 1 ijms-23-10651-f001:**
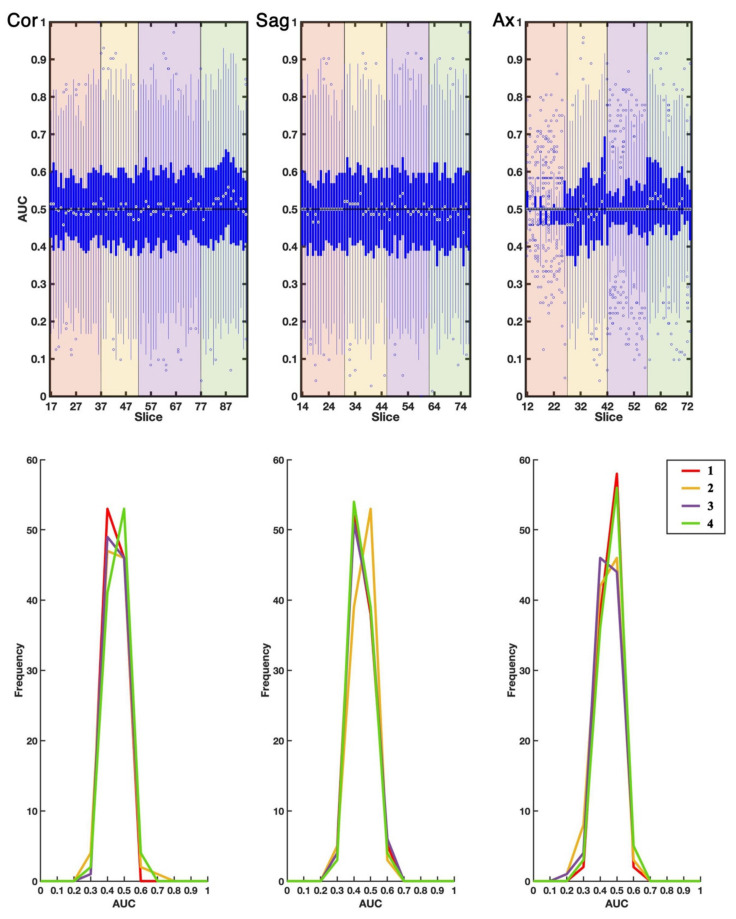
AUC statistics. First, second and third columns refer to coronal (Cor), sagittal (Sag) and axial (Ax) projections respectively. **Top**—Distribution functions of the EDSS-Ds’ AUCs in each slice. White circles represent the medians of the distributions per slice, blue thick lines define the distributions portions between the 25th and 75th percentiles, blue thin lines extend to the most extreme data points not considered outliers. Empty circles are the best-performing models (outliers). Red, yellow, purple, and green areas distinguish slabs 1 to 4. **Bottom**—AUC distribution functions displayed as histogram, derived from the predictions for folding all slices in each slab. Red, yellow, purple, and green lines correspond to slabs 1 to 4.

**Figure 2 ijms-23-10651-f002:**
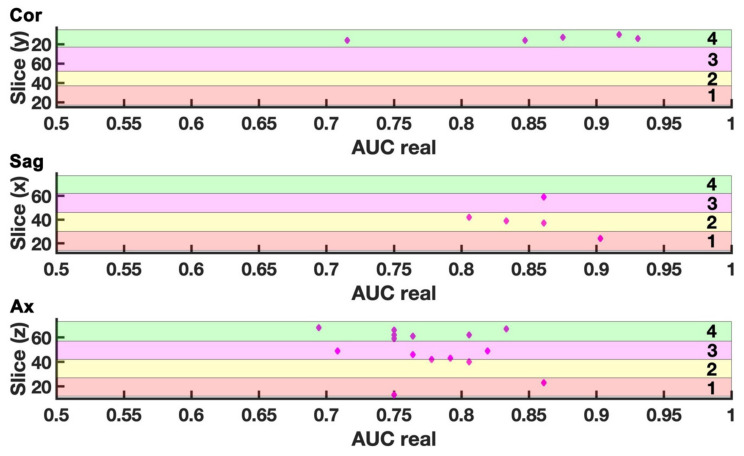
Slice-classifier analysis. Plot of AUC of the real models versus the relative location, thus the slice number, distinguishing coronal (Cor), sagittal (Sag), and axial (Ax) projections and slabs 1, 2, 3, 4. Models satisfying the double criteria of having null AUC = 0.5 and Z > 1.96 are displayed as magenta diamonds. These selected classifiers also identify brain portions as precursors of diseases worsening. Diamonds circled in black are classifiers performing well in more than 1 slice.

**Figure 3 ijms-23-10651-f003:**
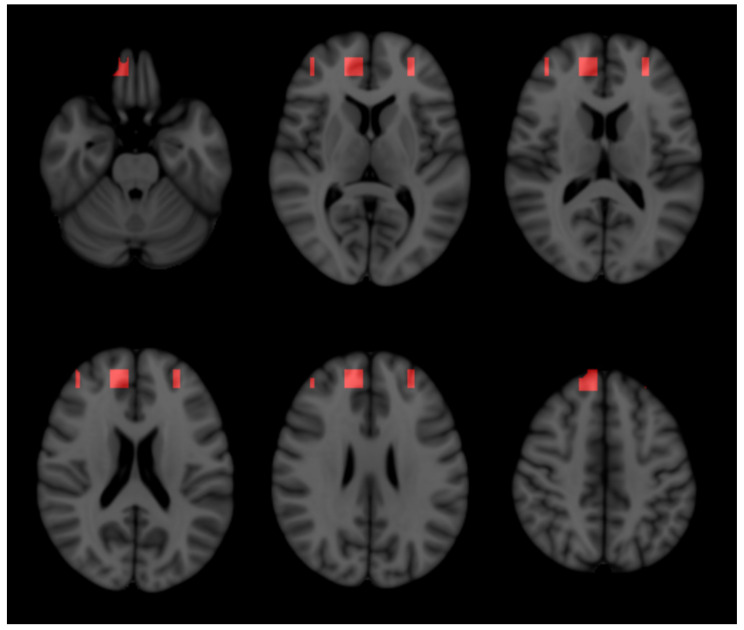
Three-dimensional map. To obtain a hint of the areas that are more informative in predicting disability progression, we intersected all those slices corresponding to at least one model fulfilling the criteria for predicting disability progression. The voxels corresponding to the intersections are superimposed onto the three-dimensional T1-weighted brain magnetic resonance image and displayed as red areas. Confluence of significant slices is denser in frontal areas, mainly within the grey matter.

**Figure 4 ijms-23-10651-f004:**
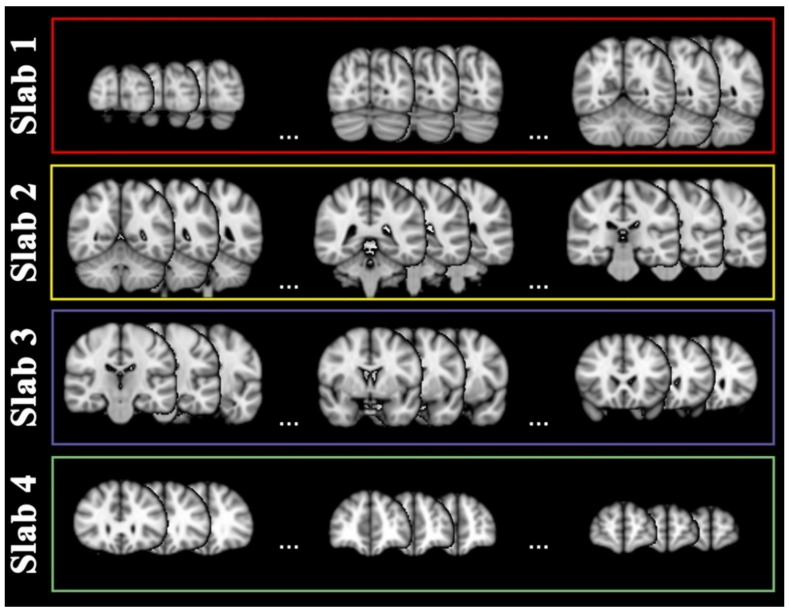
Coronal slabs. Coronal projections of magnetic resonance images were divided into four slabs: slab 1 covered slices characterizable by the coordinates y = [17–36] as sampled in the red rectangle, slab 2 by y = [37–51] as sampled in the yellow rectangle, slab 3 by y = [52–76] as sampled in the purple rectangle, slab 4 by y = [77–95] as sampled in the green rectangle. For display purposes, nine slices are shown as sampled from the limits and center of each slab range.

**Figure 5 ijms-23-10651-f005:**
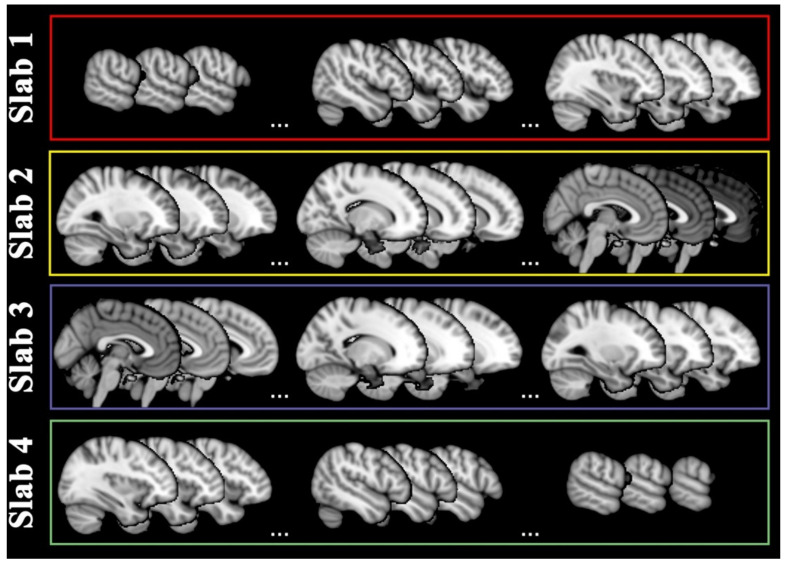
Sagittal slabs. Sagittal projections of magnetic resonance images were divided into four slabs: slab 1 covered slices characterizable by the coordinates x = [14–29] as sampled in the red rectangle, slab 2 by x = [30–45] as sampled in the yellow rectangle, slab 3 by x = [46–61] as sampled in the purple rectangle, slab 4 by x = [62–77] as sampled in the green rectangle. For display purposes, nine slices are shown as sampled from the limits and center of each slab range.

**Figure 6 ijms-23-10651-f006:**
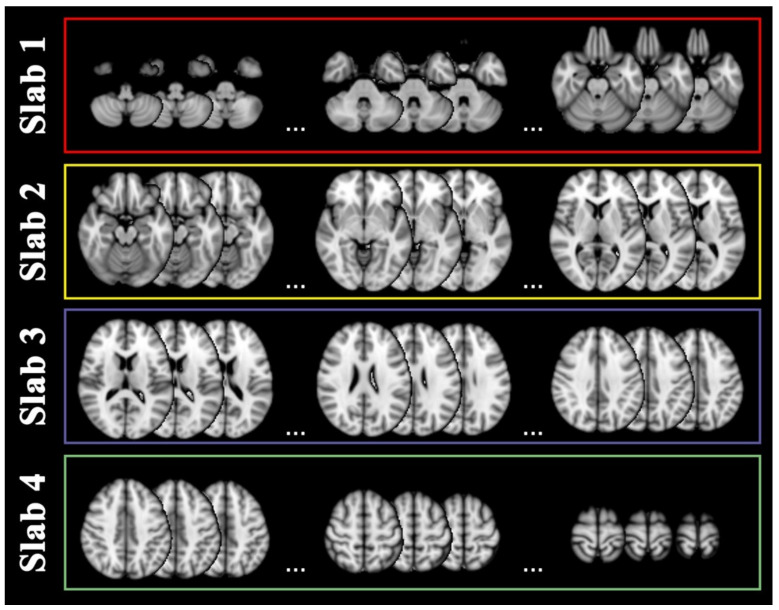
Axial slabs. Axial projections of magnetic resonance images were divided into four slabs: slab 1 covered slices characterizable by the coordinates z = [12–26] as sampled in the red rectangle, slab 2 by z = [27–41] as sampled in the yellow rectangle, slab 3 by z = [42–56] as sampled in the purple rectangle, slab 4 by z = [57–73]) as sampled in the green rectangle. For display purposes, nine slices are shown as sampled from the limits and center of each slab range.

**Figure 7 ijms-23-10651-f007:**
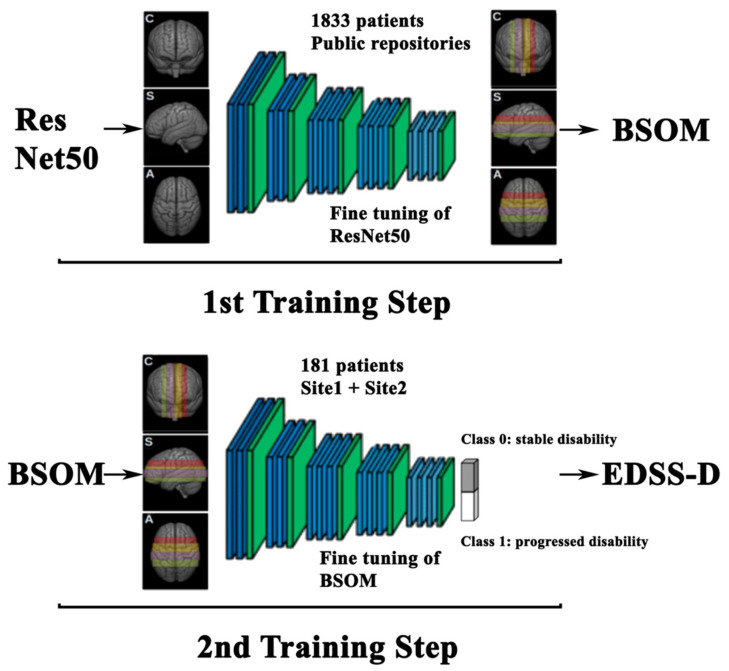
Two-step training workflow. Flowchart of the fine-tuning of the ResNet50 deep learning network including a two-step training performed on jpg-exported MRI images. Training step 1—Brain Scan Optimized Model (BSOM, top): on a sample of 1833 MRIs collected from public repositories, the network learned the brain basic structural concepts classifying generic brain three-dimensional T1-weighted magnetic resonance coronal (C), sagittal (S) and axial (A) images by location, thus into four adjacent slabs. Each slab of a projection plane is identified by a different color that will be kept in the following figures: slab 1 in red, slab 2 in yellow, slab 3 in purple, and slab 4 in green. Slab positions for coronal, sagittal, and axial projections followed the standard planes parallel to the floor of the IV ventricle, to the medial plane and to the bicommissural line, respectively. For display purposes, each slab position is shown as a rectangle covering brain portions approximately coped to the slices included in the slab itself. Training step 2—EDSS-Discriminators (EDSS-D, bottom): each BSOM was adapted to the EDSS classification problem (class 0 and class 1) of the investigated sample of patients with multiple sclerosis recruited in sites 1 and 2. Each projection model was separately trained only on the slices belonging to each slab subdividing the A, C, and S planes (different color regions). As a result, 12 independent EDSS-Discriminators were achieved. Blue and green layers schematically represent those convolutional and pooling layers that build the neural network.

**Figure 8 ijms-23-10651-f008:**
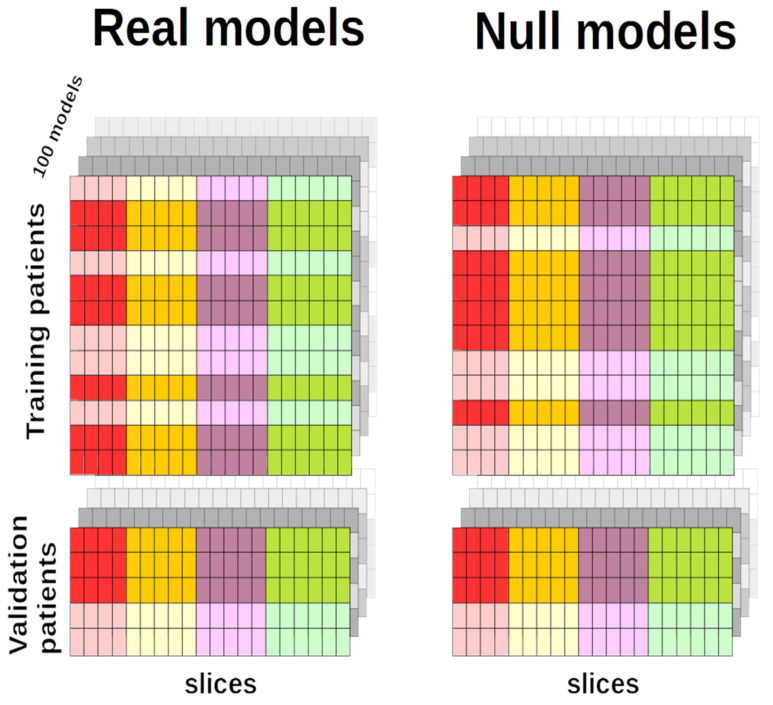
Normal vs. null classifiers. Real models—100 models were fine-tuned to classify patients’ disability progression. In this schematic representation, any sheet represents an ideal model and stands for a single training and validation set. A single entry (small rectangles) of the sheet represents an MRI image (slice) and red, yellow, purple, and green lines correspond to slabs 1 to 4. Darker entries represent patients in class 0, while light elements stand for patients in class 1. Projection-wise, an EDSS-D was trained only on scans of the same color. In total, the EDDS-Ds were 100 (number of training sets) × 4 (slabs per projection plane) × 3 (axial, coronal, or axial projections). A training set was composed of 163 patients, 107 negatives (dark rows) and 56 positives (light rows); validation consisted of 18 patients, 12 of which were negatives (dark rows) and 6 positives (light rows). Null models—Null models’ structure traced that of real models. Training sets were modified by the reshuffling of the labels characterizing the patients. Therefore, the numbers of negative (dark) and positive (light) patients, as well as the images, were the same as in the real model, only the information furnished during the training phase was incorrect. Validation sets remained unchanged.

**Figure 9 ijms-23-10651-f009:**
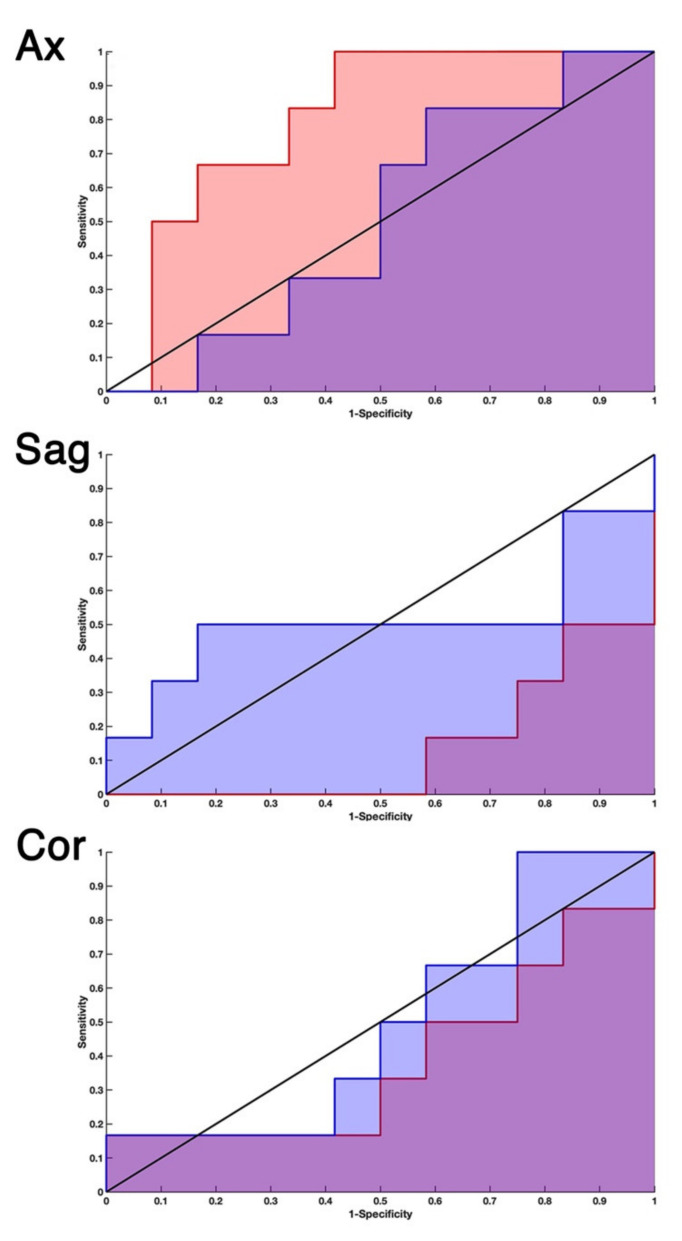
Three possible outcomes of the real vs. null AUC comparison. From the comparison of the receiver operating characteristic curves (ROCs, 1-specificity vs. sensitivity diagram) of a (real) model and its null model, we might meet three different situations. (Ax) Axial projection, slice 40. Null AUC = 0.51 and real AUC = 0.80, in this case Z > 1.96. (Sag) Sagittal projection, slice 49. Null AUC = 0.51 and real AUC = 0.14, in this case Z < −1.96. (Cor) Coronal projection, slice 64). Null AUC = 0.5 and real AUC = 0.4, in this case, no statistical difference between the two models could be assessed. The stepsize form of the ROC could be traced to the amplitude of the validation sets (12 negatives and 6 positives). The minimum increment of the area is therefore 1/(6 × 12).

**Table 1 ijms-23-10651-t001:** Demographics and Clinical data of patients.

	All Subjects (Site 1 + Site 2)	Subjects at Site 1	Subjects at Site 2	Between Sites Comparison
	Average (std)	Average (std)	Average (std)	z-(*p*-Value)
**Number**	181	105	76	-
**Age [years]**	39.57 (10.46)	38.29 (9.75)	41.33 (11.20)	−1.90(0.06)
**Sex (F/M)**	112/69	80/25	32/44	**21.71 (0.001) ***
**Phenotype (RR/P)**	136/45	85/20	51/25	**4.53 (0.04) ***
**Disease duration [years]**	9.90 (8.06)	8.27 (7.97)	12.06 (7.36)	**−3.31 (0.001)**
**EDSS at baseline**	3.0 [0.0–7.5] **	2.0 [0.0–7.5] **	3.5 [2.0–7.5] **	**−5.06 (0.001)**
**Time to follow up [years]**	3.94 (0.91)	4.2 (0.93)	3.53 (0.69)	**5.87 (0.001)**
**Therapy at baseline (1st line, 2nd line, none)**	58, 75, 48	32, 31, 42	26, 44, 6	-
**Therapy switch** **(no switch, switch to same line treatment, none to 1st line, none to 2nd line, 1st to 2nd line, 2nd to 1st line, 1st line to none, 2nd line to none)**	80, 32, 15, 14, 18, 13, 8, 1	46, 10, 13, 13, 9, 6, 7, 1	34, 22, 2, 1, 9, 7, 1, -	-
**Relapse**	0 [0–5] **	0 [0–5] **	0 [0–4] **	0.82 (0.41)
**Disability progression (Yes/No)**	62/119	36/69	26/50	0.0001 (0.99) *
**Progressed patients (%)**	34	34	34	-

z- and *p*-values are calculated via Mann–Whitney test if not stated otherwise. *: chi-square statistics; **: median [range]. Significant between-site differences are highlighted in bold font. EDSS: Expanded disability status scale; RR: relapsing-remitting; P: progressive.

## Data Availability

Data and code are available upon request. Public data were retrieved from https://adni.loni.usc.edu/ and https://www.ppmi-info.org/ (accessed on 1 April 2020).

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
