# Peer review of "Evaluation of Disability Progression in Multiple Sclerosis via Magnetic-Resonance-Based Deep Learning Techniques"

_ijms, 2022, doi:10.3390/ijms231810651_

Round 1
Reviewer 1 Report
The authors investigate whether a deep learning-based classification can predict the future disease progression of multiple sclerosis from 3D T1-weighted MR images covering the whole brain. Patients were imaged at baseline and at a follow-up 2 to 6 years later. A ROC-analysis of was used to identify the best classifiers on sagittal, coronal, and axial images. Axial images at the frontal lobe were found to be the most informative for predicting disease progression.
The manuscript describes an advanced method for prognosis of disease progression, using a ResNet50 deep learning network. This is an interesting study on an important topic. The number of training subjects was relatively low, so this study may be regarded as a pilot project towards an automated classification. The manuscript could be improved in terms of more clearly demonstrating the ability and potential of the method to discriminate progressing individuals from non-progressing. Since the training process of the deep learning network is quite complex, the description of this part could be more detailed and clear, see detailed comments below.
Specific points
1. References 5, 13, 14, 15, 37, and possibly others, are incomplete. Please format all references equally, with authors, title, journal, year, volume and page range.
2. L58: Reference 5 is not relevant for visual recognition.
3. L60: It is excessive with 10 references on image segmentation. Please reduce, and instead add more references on MS disease progression and deep learning (see next point).
4. More papers, not cited in the manuscript, have used deep learning for prediction of MS progression:
- Storelli L et al. A Deep Learning Approach to Predicting Disease Progression in Multiple Sclerosis Using Magnetic Resonance Imaging. Invest Radiol. 2022 Jul 1;57(7):423-432. PMID: 35093968.
- Peng Y et al. Prediction of unenhanced lesion evolution in multiple sclerosis using radiomics-based models: a machine learning approach. Mult Scler Relat Disord. 2021 Aug;53:102989. PMID: 34052741.
Also other non-MRI studies can be mentioned:
- De Brouwer E et al. Longitudinal machine learning modeling of MS patient trajectories improves predictions of disability progression. Comput Methods Programs Biomed. 2021 Sep;208:106180. PMID: 34146771.
- Law MT et al. Machine learning in secondary progressive multiple sclerosis: an improved predictive model for short-term disability progression. Mult Scler J Exp Transl Clin. 2019 Nov 6;5(4):2055217319885983. PMID: 31723436.
5. Table 1 lists several characteristics of the patients, e.g. disease duration, EDSS at baseline, therapy switch. Were any of these parameters correlated to the ability of the DL network to make a correct/incorrect prediction?
6. L117: Please also state the inversion time (TI).
7. L121: What is meant by “corrected for field inhomogeneity”? Does this mean a correction of spatially varying image intensity?
8. L127-128 and L206-207: were the slices oblique or not? In figure 4, the slabs appear to be non-oblique, but how is this “not as in reality”?
9. L179: I assume the images in the first training step were also in MNI space? This could be explicitly stated.
10. In figure 4 legend: what is meant by left and right? (BSOM, left), (EDSS-D, right). Please also explain the difference between blue and green layers.
11. L229: Please show example images before and after applying the homemade code to maximize contrast.
12. L235-236: I have trouble understanding “Having established the validation dataset amplitude as the 10% of the images in each projection plane,…”.
13. L248-249: “Each training set included 90% of the total patients: 163 patients subdivided into 107 negatives and 56 positives.” Is this equivalent to a 10-fold cross-validation?
14. I also have a problem with understanding the nomenclature in the figure 5 legend. Is a “sheet” same as a “model” (and in turn same as one training+validation data set)?
15. L321 (and throughout the manuscript): the notation AUC<|0.5+delta| does not make sense. I assume it should be |AUC-0.5|<delta?
16. Results and Discussion: The authors have chosen to present AUC-values, rather than classification accuracy. I do not argue against that, but I nevertheless suggest that also the classification accuracy for the validation patients should be stated. Otherwise, it is quite difficult for the reader to understand the potential of the method, and it is difficult to assess the proposed method in relation to other published methods. Preferably, the authors should attempt to briefly discuss the performance of the suggested method relative the methods in references 25, 27, and 28.
Minor points
17. Table 1 (therapy switch): “lime” should be “line”.
18. L225: “built-in”.
19. Please label the three graphs in figure 8 with “Cor”, “Sag” and “Ax” for clarity.
Author Response
The authors investigate whether a deep learning-based classification can predict the future disease progression of multiple sclerosis from 3D T1-weighted MR images covering the whole brain. Patients were imaged at baseline and at a follow-up 2 to 6 years later. A ROC-analysis of was used to identify the best classifiers on sagittal, coronal, and axial images. Axial images at the frontal lobe were found to be the most informative for predicting disease progression.
The manuscript describes an advanced method for prognosis of disease progression, using a ResNet50 deep learning network. This is an interesting study on an important topic. The number of training subjects was relatively low, so this study may be regarded as a pilot project towards an automated classification. The manuscript could be improved in terms of more clearly demonstrating the ability and potential of the method to discriminate progressing individuals from non-progressing. Since the training process of the deep learning network is quite complex, the description of this part could be more detailed and clear, see detailed comments below.
Specific points
- References 5, 13, 14, 15, 37, and possibly others, are incomplete. Please format all references equally, with authors, title, journal, year, volume and page range.
We thank the referee for having noticed how incomplete were some references and carefully checked all of them.
- L58: Reference 5 is not relevant for visual recognition.
We modified the reference to a paper by Russakovsty et al. 2015, who provided a detailed analysis of large-scale image classification and object detection. Russakovsky, Olga, et al. "Imagenet large scale visual recognition challenge." International journal of computer vision 115.3 (2015): 211-252.
- L60: It is excessive with 10 references on image segmentation. Please reduce, and instead add more references on MS disease progression and deep learning (see next point).
See point below, please.
- More papers, not cited in the manuscript, have used deep learning for prediction of MS progression:
- Storelli L et al. A Deep Learning Approach to Predicting Disease Progression in Multiple Sclerosis Using Magnetic Resonance Imaging. Invest Radiol. 2022 Jul 1;57(7):423-432. PMID: 35093968.
- Peng Y et al. Prediction of unenhanced lesion evolution in multiple sclerosis using radiomics-based models: a machine learning approach. Mult Scler Relat Disord. 2021 Aug;53:102989. PMID: 34052741.
Also other non-MRI studies can be mentioned:
- De Brouwer E et al. Longitudinal machine learning modeling of MS patient trajectories improves predictions of disability progression. Comput Methods Programs Biomed. 2021 Sep;208:106180. PMID: 34146771.
- Law MT et al. Machine learning in secondary progressive multiple sclerosis: an improved predictive model for short-term disability progression. Mult Scler J Exp Transl Clin. 2019 Nov 6;5(4):2055217319885983. PMID: 31723436.
Following referee’s suggestion, we reduced the number of citations of papers referring to brain lesion and structure segmentation, keeping one reference for each topic (e.g., brainstem segmentation, enhancing lesion recognition, and so on). Further, we included references to papers using artificial intelligence to predict clinical status and worsening in patients with MS (L66-68): “Indeed, to predict clinical status and worsening in patients with MS, artificial intelligence algorithms were applied either on MRI [15–18] and/or clinical data [19,20]”.
- Table 1 lists several characteristics of the patients, e.g. disease duration, EDSS at baseline, therapy switch. Were any of these parameters correlated to the ability of the DL network to make a correct/incorrect prediction?
As expressed by the referee, all the cited characteristics of patients may be correlated with disability progression and influence the classification models. Indeed, we investigated this topic in another work of ours, that was published in Tommasin et al. 2021 and cited in this manuscript. Certainly, it would be of interest to subsample the subjects recruited in this study in order to study how the same characteristics influence the DL network, but the sample size would not allow such a further division.
- L117: Please also state the inversion time (TI).
As suggested by the referee, we added the TI, that was 900ms for both sites.
- L121: What is meant by “corrected for field inhomogeneity”? Does this mean a correction of spatially varying image intensity?
Correct, we clarified this point in L128-129: “3DT1 images were corrected for field inhomogeneity, thus spatial variations of image intensity, brain-extracted and registered onto standard space via linear registration, as implemented in the FMRIB Software Library v6.0 (https://fsl.fmrib.ox.ac.uk/fsl/fslwiki).”
- L127-128 and L206-207: were the slices oblique or not? In figure 4, the slabs appear to be non-oblique, but how is this “not as in reality”?
Slices were cut as explained in the cited lines, thus along coronal, sagittal and axial projections following the standard planes parallel to the floor of the IV ventricle, to the medial plane and to the bicommissural line. In figure 4 caption, for each plane, we showed the 4 slabs created by grouping the slices just for illustration purpose and “not in reality”, thus not identifying the right brain portions covered by each slab. Exact brain portions for each plan and slab are shown in figures 1, 2 and 3. We changed the sentence to better clarify how slabs are displayed in L218-224: “For display purposes, each slab position is shown as a rectangle covering brain portions ap-proximately coped to the slices included in the slab itself”.
- L179: I assume the images in the first training step were also in MNI space? This could be explicitly stated.
Correct, we explicitly stated this point in L189-192: “Preprocessing of the public repository images was performed following the same procedure as the study images, thus images were corrected for field inhomogeneity, brain-extracted and registered onto MNI space via linear registration.”
- In figure 4 legend: what is meant by left and right? (BSOM, left), (EDSS-D, right). Please also explain the difference between blue and green layers.
We are sorry for the typos, we corrected the label of the two images that are placed at the top and the bottom of the figure, instead of the left and right (L211-212 and L224): “Training step 1 – Brain Scan Optimized Model (BSOM, top)”, “Training step 2 – EDSS-Discriminators (EDSS-D, bottom)”. As we clarified in the text, blue and green layers schematically represent those convolutional and pooling layers that build the neural network. We clarified it in the figure caption L228-229.
- L229: Please show example images before and after applying the homemade code to maximize contrast.
In order to follow referee’s suggestion, we included a Supplementary Material document including two examples of optimized slices extracted from the MR image of a subject.
- L235-236: I have trouble understanding “Having established the validation dataset amplitude as the 10% of the images in each projection plane,…”.
We rephrased to better explain how we devised in training and validation datasets (L253-255): “To evaluate models’ confusion matrix, the images distinguished in the three planes, e.g., coronal, sagittal and axial, were devised in training and validation sets as the 90% and the 10% for each plan.”
- L248-249: “Each training set included 90% of the total patients: 163 patients subdivided into 107 negatives and 56 positives.” Is this equivalent to a 10-fold cross-validation?
Correct.
- I also have a problem with understanding the nomenclature in the figure 5 legend. Is a “sheet” same as a “model” (and in turn same as one training+validation data set)?
Correct, we clarified it in figure 5 caption (L294-295): “In this schematic representation any sheet represents an ideal model and stands for a single training and validation set.”
- L321 (and throughout the manuscript): the notation AUC<|0.5+delta| does not make sense. I assume it should be |AUC-0.5|<delta?
We corrected the notation to mean that the valid null classifiers were those classifiers whose AUC was included in the range delta +/-0.5: AUCÎ[0.5±?].
- Results and Discussion: The authors have chosen to present AUC-values, rather than classification accuracy. I do not argue against that, but I nevertheless suggest that also the classification accuracy for the validation patients should be stated. Otherwise, it is quite difficult for the reader to understand the potential of the method, and it is difficult to assess the proposed method in relation to other published methods. Preferably, the authors should attempt to briefly discuss the performance of the suggested method relative the methods in references 25, 27, and 28.
Following referee’s comment, we added a brief discussion regarding the benefit of using AUC vs accuracy to test the discriminative power of our learning strategy (L310-314): “AUC is in fact often preferred over accuracy for binary classification on highly unbalanced problem. In our specific case, indeed, negative patients are in average 65% of the total both in the training and validation sets, yielding an accuracy of 0.7 for any model. Therefore, models appeared to be representative but not discriminative, suggesting that we were overfitting to a single class, namely the negative class of patients.”
Minor points
- Table 1 (therapy switch): “lime” should be “line”.
- L225: “built-in”.
- Please label the three graphs in figure 8 with “Cor”, “Sag” and “Ax” for clarity.
We corrected these points following referee’s suggestion.
Reviewer 2 Report
This manuscript describes an opportunity for better prediction of clinical course in patients with multiple sclerosis as a brain structure-based deep learning method using clinical magnetic resonance images (MRI), which have a potential for improving the clinical practices noninvasively.
The contents are good enough to provide information in patient demographics and each method comprehensively in both descriptive and visually guided manner. The main topics are well discussed in paragraph writing; the hypothesis and conclusion is consistent.
Therefore, I think the manuscript is acceptable except a minor point as shown below,
1) In this study, the authors utilized T1-weighted MRI. On the other hand, T2-weighted and proton-weighted MRIs are also quite informative. The authors should comment the reason why the T2wMRIs were not utilized and show some potential of using other contrast images.
I think that the authors could reconsider these points to make the manuscript relevant for interested readers and show how it can be clinically useful.
I hope you may kindly consider my suggestion. Thank you.
Author Response
This manuscript describes an opportunity for better prediction of clinical course in patients with multiple sclerosis as a brain structure-based deep learning method using clinical magnetic resonance images (MRI), which have a potential for improving the clinical practices noninvasively.
The contents are good enough to provide information in patient demographics and each method comprehensively in both descriptive and visually guided manner. The main topics are well discussed in paragraph writing; the hypothesis and conclusion is consistent.
Therefore, I think the manuscript is acceptable except a minor point as shown below,
1) In this study, the authors utilized T1-weighted MRI. On the other hand, T2-weighted and proton-weighted MRIs are also quite informative. The authors should comment the reason why the T2wMRIs were not utilized and show some potential of using other contrast images.
I think that the authors could reconsider these points to make the manuscript relevant for interested readers and show how it can be clinically useful.
I hope you may kindly consider my suggestion. Thank you.
We agree with the referee and think that the use of T2-weighted images will improve the performance of the models. Indeed, we included this as a limit point in the discussion (L562-567) and we are working on the development of the models including T2-weighted and contrast-enhanced images. However, at the moment, we do not have such a sample size to be relevant for a study.